# NeuroKoop: Neural Koopman Fusion of Structural–Functional Connectomes for Identifying Prenatal Drug Exposure in Adolescents

Badhan Mazumder[1], Aline Kotoski[2], Vince D. Calhoun[3], Dong Hye Ye[4]

[1,4]*Department of Computer Science, Georgia State University*
[2]*Neuroscience Institute, Georgia State University*
[1,2,3,4]*Tri-Institutional Center for Translational Research in Neuroimaging and Data Science (TReNDS),*
*Georgia State University, Georgia Institute of Technology, and Emory University*
Atlanta, GA, USA
bmazumder1@gsu.edu[1], akotoski1@gsu.edu[2], vcalhoun@gsu.edu[3], dongye@gsu.edu[4]

*Abstract*—**Understanding how prenatal exposure to psychoactive substances such as cannabis shapes adolescent brain organization remains a critical challenge, complicated by the complexity of multimodal neuroimaging data and the limitations of conventional analytic methods. Existing approaches often fail to fully capture the complementary features embedded within structural and functional connectomes, constraining both biological insight and predictive performance. To address this, we introduced NeuroKoop, a novel graph neural network-based framework that integrates structural and functional brain networks utilizing neural Koopman operator-driven latent space fusion. By leveraging Koopman theory, NeuroKoop unifies node embeddings derived from source-based morphometry (SBM) and functional network connectivity (FNC) based brain graphs, resulting in enhanced representation learning and more robust classification of prenatal drug exposure (PDE) status. Applied to a large adolescent cohort from the ABCD dataset, NeuroKoop outperformed relevant baselines and revealed salient structural–functional connections, advancing our understanding of the neurodevelopmental impact of PDE.**

*Index Terms*—**Prenatal Drug Exposure, Structural–Functional Fusion, Neural Koopman Operator, Graph Neural Network**

## I. INTRODUCTION

Adolescence marks a critical stage of brain development, characterized by extensive remodeling of neural circuits that support cognition, behavior, and emotional regulation [1], [2]. Accumulating evidence indicates that prenatal exposure to psychoactive substances—particularly cannabis—can contribute to lasting alterations in brain connectivity and neurocognitive functioning [3], [4]. As cannabis use became increasingly common during pregnancy nowadays, clarifying its potential impact on neurodevelopment is both a scientific and public health priority. Although previous studies [4], [5] have demonstrated that prenatal cannabis exposure may lead to enduring changes in neural organization, the precise mechanisms and specific brain network alterations involved remain poorly understood, hindering opportunities for timely intervention and support.

Recent advances in multimodal neuroimaging [6] provide a unique window into the architecture of the adolescent brain. Structural measures, such as source-based morphometry (SBM) derived from structural magnetic resonance imaging (sMRI), captures inter-network gray matter covariation, while functional network connectivity (FNC) from resting-state fMRI reveals the temporal synchrony of distributed brain networks. The joint analysis of structural and functional connectomes holds promise for uncovering subtle neurobiological effects of early exposures that may be invisible to unimodal analysis. However, several key obstacles remain. First, most existing analytic frameworks [7]–[12] either examine these modalities in isolation or perform naïve integration by simple feature concatenation, neglecting the complex, nonlinear dependencies and cross-modal dynamics inherent in brain networks. Second, standard deep learning approaches [13], [14], including many recently proposed graph neural network (GNN)-based frameworks [9]–[12], [15], tend to focus on local connectivity patterns while overlooking higher-order and dynamic interactions across brain regions. This limits their capacity to model the complex structure-function interaction and cognitive modulation effects that are critical for understanding the neural impact of prenatal exposures. Third, most studies [10], [16] frequently overlook important sources of cognitive heterogeneity, such as individual differences in working memory (WM), which are not only crucial determinants of adolescent outcomes but may themselves be subtly altered by prenatal drug exposure (PDE), as suggested by prior studies [5], [17]. Neglecting to model these individual differences risks introducing confounding effects and limits the biological interpretability of findings, potentially masking key mechanisms by which neurodevelopmental perturbations exert their influence.

To address these gaps, we propose NeuroKoop, a novel GNN based multimodal framework that fuses structural and functional connectomes via a neural Koopman operator-guided latent dynamics fusion. Inspired by the theoretical strengths of Koopman operator theory [18]—which provides a linear yet expressive mapping for analyzing complex, nonlinear dynamical systems—our approach projects both structural and

functional graphs into a shared latent space. Here, cross-modal information was exchanged and refined, enabling the model to learn richer and more biologically plausible representations of brain organization. Additionally, we incorporated WM scores as subject-specific conditioning signals within the fusion process. WM was selected based on its well-established role as a core cognitive domain affected by PDE, with prior studies [5], [17] reporting both behavioral deficits and disruptions in brain network organization among exposed individuals. From a developmental neuroscience perspective, WM is considered closely tied to the maturation of large-scale brain systems particularly those supporting executive function and self-regulation, and is known to be heritable, well-characterized in adolescence, and strongly associated with distributed patterns of functional connectivity [19], [20]. By integrating WM scores into the latent fusion process, NeuroKoop tries to disentangle exposure-specific effects from broader individual variability, thereby enhancing both predictive performance and underlying neuroscientific insights.

In brief, our contributions can be outlined as follows:

- We proposed a novel GNN framework that aligned and integrated structural and functional brain networks through neural Koopman operator, enabling a unified latent representation.
- We incorporated individual WM measures as auxiliary information, enhancing the robustness of network fusion by leveraging cognitive mechanisms underlying exposure effects.
- Extensive evaluation on a large adolescent cohort demonstrated NeuroKoop's superiority over state-of-the-art (SOTA) fusion baselines.

## II. METHODOLOGY

Let $\mathcal{D} = \{(\mathbf{A}_i, \mathbf{B}_i), c_i, t_i\}_{i=1}^N$ denote a dataset of $N$ subjects, where $\mathbf{A}_i \in \mathbb{R}^{Q \times Q}$ and $\mathbf{B}_i \in \mathbb{R}^{Q \times Q}$ are the structural and functional connectivity matrices for the $i$-th subject, each defined over $Q$ brain networks. Here, $c_i \in \mathbb{R}$ is the individual WM score, and $t_i \in \{0, 1\}$ indicates PDE status. Subject-specific SBM matrices $\mathbf{A}_i$ were constructed as the outer product of the subject's SBM loading vector, $\mathbf{A}_i = \mathbf{l}_i \mathbf{l}_i^\top$, with $\mathbf{l}_i \in \mathbb{R}^\eta$ representing their structural features across $\eta$ SBM components. As illustrated in Figure 1, our goal is to build a predictive framework that leverages multimodal connectomes and cognitive assessment to determine each individual's PDE status $\hat{t}_i$.

### A. Modality-Wise Graph Encoding

For the given dataset $\mathcal{D}$, we constructed for each subject a structural graph $\mathcal{G}_i^S = (\mathcal{V}, \mathcal{E}_i^S, \mathbf{X}_i^S)$ and a functional graph $\mathcal{G}_i^F = (\mathcal{V}, \mathcal{E}_i^F, \mathbf{X}_i^F)$. Here, $\mathcal{V}$ represents the shared set of $Q$ brain networks, while subject-specific edges $\mathcal{E}_i^S$ and $\mathcal{E}_i^F$ are defined by applying $k$-nearest neighbor (kNN) sparsification to the respective SBM ($\mathbf{A}_i$) and FNC ($\mathbf{B}_i$) matrices. Node feature matrices $\mathbf{X}_i^S$ and $\mathbf{X}_i^F$ were constructed such that each node's feature vector corresponds to the respective row of $\mathbf{A}_i$ and $\mathbf{B}_i$, respectively.

Modality-specific GNN encoders, $\mathcal{F}_S$ and $\mathcal{F}_F$, were then applied to the full sets of structural and functional graphs $\mathcal{G}^S = \{\mathcal{G}_i^S\}_{i=1}^N$ and $\mathcal{G}^F = \{\mathcal{G}_i^F\}_{i=1}^N$ to obtain latent node representations:

$$\mathbf{H}^S = \mathcal{F}_S(\mathcal{G}^S), \qquad \mathbf{H}^F = \mathcal{F}_F(\mathcal{G}^F) \qquad (1)$$

where $\mathbf{H}^S, \mathbf{H}^F \in \mathbb{R}^{N \times Q \times d}$ denote the collections of latent node embeddings for the structural and functional modalities, and $d$ is the embedding dimension. We employed traditional graph convolutional networks (GCNs) [21] as our encoders due to their effectiveness for simply capturing topological patterns in brain graphs [11] and their robustness for subject-level representation learning.

### B. Neural Koopman-Driven Multimodal Fusion

*1) Dynamic Latent Fusion via Neural Koopman Operator:* To robustly integrate structural and functional connectomes, after deriving node-level latent representations from both modalities we employed a bidirectional cross-modal attention layer (CAL) [22] that allowed each modality to attend to the other. Specifically, we applied two parallel scaled dot-product attention operations: one where the structural embedding ($\mathbf{H}^S$) served as the query ($\mathbf{Q}$) and the functional embedding ($\mathbf{H}^F$) provided the keys ($\mathbf{K}$) and values ($\mathbf{V}$), and a second where this relationship was reversed. This design facilitated the exchange of information between modalities and accentuates salient inter-modality relationship. For each node, the resulting attention-weighted features from both directions were concatenated, yielding an initial fused representation $\mathbf{Z}_0 \in \mathbb{R}^{Q \times 2d}$. This joint embedding preserved both intra- and inter-modality topological patterns and provides a unified basis for subsequent dynamic latent fusion via the neural Koopman operator.

To move beyond static fusion, we introduced a neural Koopman operator that dynamically evolved the fused latent representation for each subject. Drawing inspiration from Koopman operator theory [18]—originally developed for dynamical systems—we employed a learnable neural operator [23] to iteratively refine the fused node states in a cognitively informed manner. Unlike classic Koopman approaches, which operated on temporal sequences, we leveraged the operator's capacity to simulate virtual trajectories within the latent space, even with static connectome inputs.

Formally, for each subject, we initialized with a fused embedding $\mathbf{Z}_0$ and unrolled it through $T$ steps according to:

$$\mathbf{Z}_{t+1} = U_\psi(\mathbf{Z}_t) \odot M_\psi(c), \qquad t = 0, \ldots, T-1, \quad (2)$$

where $U_\psi$ and $M_\psi$ are independent multilayer perceptrons (MLPs) applied to the current latent state $\mathbf{Z}_t$ and the subject's WM score $c$, respectively, and $\odot$ denotes element-wise multiplication. While the WM score $c$ is a single scalar value, the network learns to project it into a higher-dimensional vector through $M_\psi(c)$. This enables the Koopman evolution to adapt based on subject-specific cognitive context, allowing meaningful modulation of the latent trajectory using minimal but informative input.

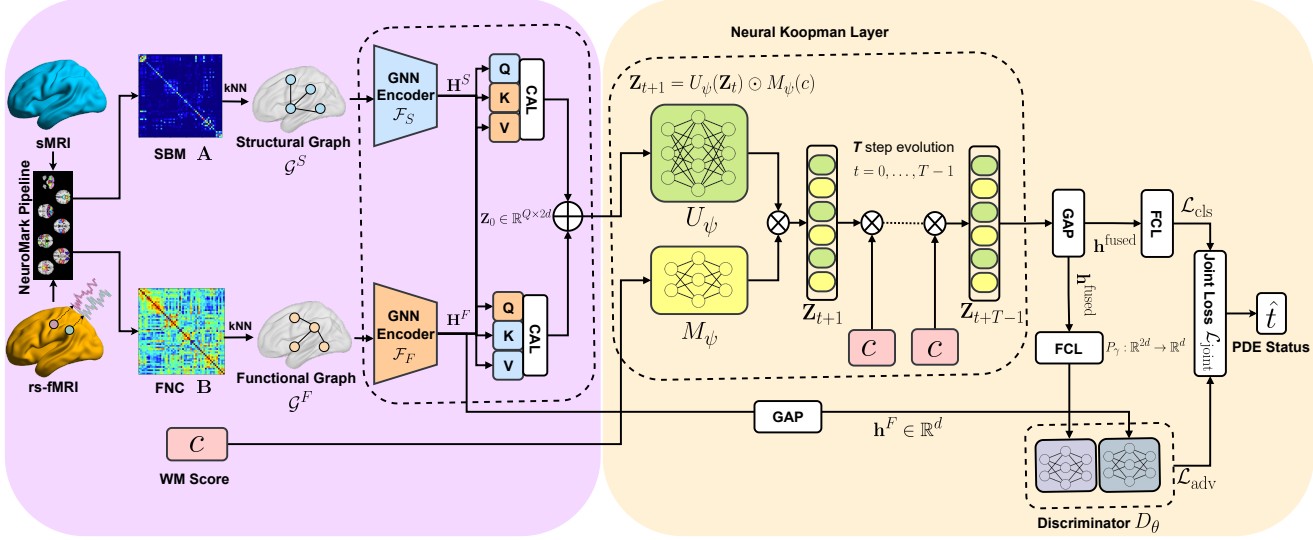

Fig. 1. NeuroKoop overview: After obtaining SBM form sMRI and FNC from rs-fMRI, modality-specific GNN encoders generated node embeddings from corresponding structural and functional brain graphs, which were fused through cross-modal attention layer (CAL). The neural Koopman layer then dynamically refined the fused latent representation based on each individual's working memory (WM) score, followed by global average pooling (GAP) to yield a subject-level vector for classifying prenatal drug exposure (PDE) status, while adversarial regularization encourages alignment with functional organization.

Through this iterative process, each subject's fused latent state was dynamically adjusted to reflect both their multimodal connectome and individual cognitive profile. This mechanism captured complex relationships between brain connectivity and cognition, allowing individualized adaptations linked to PDE and WM. Unlike conventional fusion strategies that produce static joint embeddings, our neural Koopman approach personalizes and refines the latent representation, thereby improving both predictive power and neurological relevance.

*2) Joint Training Objective and PDE Classification:* The fused and dynamically refined latent representations obtained from the neural Koopman operator, denoted as $\mathbf{Z}_T \in \mathbb{R}^{Q \times 2d}$ after $T$ Koopman steps, were pooled at the graph level and passed to a final classifier to predict the exposure status for each subject. Specifically, we aggregated node-level embeddings via global average pooling (GAP) to obtain a subject-level vector, $\mathbf{h}^{\text{fused}} = \text{GAP}(\mathbf{Z}_T) \in \mathbb{R}^{2d}$. A fully connected layer (FCL) classification head, parameterized by weights $\mathbf{w} \in \mathbb{R}^{2d}$ and bias $b \in \mathbb{R}$, was then applied to compute the exposure probability: $\hat{t} = \sigma(\mathbf{w}^\top \mathbf{h}^{\text{fused}} + b)$, where $\sigma(\cdot)$ denotes the sigmoid function.

The model was trained end-to-end with a dual-objective loss function that jointly optimized exposure classification and structural–functional representation alignment. The primary objective is binary cross-entropy loss for classification as follows:

$$\mathcal{L}_{\text{cls}} = -\frac{1}{N} \sum_{i=1}^{N} \left[ t_i \log(\hat{t}_i) + (1 - t_i) \log(1 - \hat{t}_i) \right], \quad (3)$$

where $t_i$ is the true exposure label and $\hat{t}_i$ is the predicted probability for subject $i$.

To further encourage the fused latent space to capture meaningful structure–function relationships, we incorporated an adversarial loss that distinguished between true FNC patterns and those obtained via our fusion model. Specifically, we employed a discriminator network $D_\theta$, implemented with two multilayer perceptrons (MLPs), which received as input the global pooled embedding from the FNC GNN encoder (as real) and the projected pooled embedding from the Koopman fusion branch (as synthetic), and was trained to distinguish between the two sources. By adversarially aligning the distributions of the Koopman-fused and base FNC embeddings, this mechanism acts as a functional regularizer—encouraging the fused space to remain anchored to biologically meaningful functional organization, promoting more neurologically grounded representation and reducing the risk of implausible cross-modal fusion artifacts.

Since the Koopman-fused representation $\mathbf{h}^{\text{fused}} \in \mathbb{R}^{2d}$ and the pooled FNC embedding $\mathbf{h}^F = \text{GAP}(\mathbf{H}^F) \in \mathbb{R}^d$ differ in dimension, we employed a linear projection layer $P_\gamma : \mathbb{R}^{2d} \to \mathbb{R}^d$ to map the fused vector to the same space as the FNC representation before input to the discriminator. Thus, the adversarial loss is formulated as:

$$\mathcal{L}_{\text{adv}} = \frac{1}{2N} \sum_{i=1}^{N} \left[ \log D_\theta(\mathbf{h}_i^F) + \log \left( 1 - D_\theta(P_\gamma(\mathbf{h}_i^{\text{fused}})) \right) \right],$$
$$(4)$$

where $P_\gamma(\cdot)$ denotes the linear projection of the Koopman-fused pooled embedding.

The overall training objective combines the binary cross-entropy classification loss with the adversarial loss:

$$\mathcal{L}_{\text{joint}} = \mathcal{L}_{\text{cls}} + \lambda_{\text{adv}} \mathcal{L}_{\text{adv}}, \quad (5)$$

where $\lambda_{\text{adv}}$ controls the strength of adversarial regularization.

This dual-objective optimization encourages the network not only to maximize PDE classification performance, but also to ensure that the fused representations remain grounded by the true functional brain dynamics, thereby enhancing robustness, generalization, and neurological plausibility.

## III. Results and Discussion

### A. Dataset and Data Pre-processing

The Adolescent Brain Cognitive Development (ABCD) study [6] is a comprehensive, multi-site longitudinal research initiative designed to elucidate how diverse biological, environmental, and social influences shape cognitive and mental health trajectories from childhood through adolescence across the United States. For our analysis, we leveraged the baseline ABCD dataset comprising 7,289 children aged 9 to 10 years, each with available rs-fMRI, sMRI, and WM assessments. Among these, 430 participants were identified as having prenatal cannabis exposure. To ensure a balanced comparison, we subsampled 430 non-exposed controls from the remaining 6,859 non-exposed participants, resulting in a final cohort of 860 individuals. To reduce site-specific variability, all imaging features were derived using Neuromark's standardized ICA pipeline, which ensures spatial consistency across acquisition sites. Only subjects with complete and high-quality sMRI, rs-fMRI, and WM scores at baseline were retained for analysis. Subjects with missing data or failing Neuromark QC [26] were excluded during preprocessing.

To derive subject-level functional connectivity profiles, we utilized the *Neuromark* [26] framework, which implements a spatially constrained independent component analysis (ICA) approach. Specifically, for each individual, adaptive-ICA was first used to extract 53 reproducible intrinsic connectivity networks (ICNs) along with their associated time courses. Pairwise Pearson correlations among these time courses produced individualized $53 \times 53$ functional network connectivity (FNC) matrices, representing the functional connectome for each subject.

Structural features were extracted from sMRI scans using the same *Neuromark* [26] pipeline within the GIFT toolbox. A constrained ICA was performed based on the standardized 53-component template, yielding source-based morphometry (SBM) [27], [28] loading parameters that capture independent spatial patterns of gray matter volume (GMV) variation across the cohort. Each participant's structural brain organization was summarized as a 53-dimensional vector of SBM loadings, reflecting their individual expression of distributed GMV patterns.

### B. Experimental Settings

All model development and experimentation were conducted using the PyTorch framework and ran on an NVIDIA V100 GPU. To identify robust hyperparameters for NeuroKoop, we carried out a comprehensive grid search across batch sizes: $8, 16, 32, 64, 128$, learning rates:

$1 \times 10^{-3}, 1 \times 10^{-4}, 3 \times 10^{-4}, 5 \times 10^{-4}$, GNN latent dimensions $32, 64, 128$, Koopman operator steps $1, 3, 5, 7$, and weight decay values $1 \times 10^{-5}, 5 \times 10^{-5}$. Both diagonal and full linear forms of the Koopman operator were evaluated. To sparsify the brain graphs, we applied k-nearest neighbor (k-NN) thresholding with $k = 5$ to both structural and functional modalities matrices. This value was selected based on thorough investigation with $k \in \{3, 5, 7, 10\}$, which showed that performance peaked at $k = 5$ and declined after that for larger values. To maintain consistency and simplicity throughout the pipeline, we reported all results in this study using $k = 5$. Additionally, all WM scores were standardized prior to model input.

Experimental evaluation relied on stratified 5-fold cross-validation, preserving an approximate $80 : 20$ train-test split within each fold. Our final selected configuration consisted of a batch size of $16$, learning rate $3 \times 10^{-4}$, hidden and latent dimension of $64$, 5 Koopman steps, $\lambda_{\text{adv}}$ value $0.2$ and weight decay of $1 \times 10^{-5}$. The whole model was trained end-to-end for $100$ epochs using the Adam optimizer.

Additionally, to ensure a fair comparison, all reported baseline models were trained using the same data splits, optimizer, and number of training epochs. Hyperparameters for each baselines were tuned individually through grid search to reflect their optimal configurations.

### C. Quantitative Evaluation

*1) Comparative Evaluation with SOTA Methods:* We benchmarked our NeuroKoop framework against two different classical classifiers and a suite of relevant multimodal GNN-based fusion baselines, all utilizing multimodal connectome data. As summarized in Table I, all methods were evaluated under an identical cross-validation protocol.

These two classical classifiers: logistic regression and random forest were trained on 106 dimensional feature vectors constructed by concatenating global average pooled SBM and FNC representations (53 features each). As shown in Table I, these shallow models, which rely on simplified summary statistics and lack spatial or cross-modal modeling capabilities, achieved substantially lower performance with higher standard deviation across folds compared to NeuroKoop.

For the GAT [24] and Graph Transformer [25] as baselines, modality-specific features were extracted in parallel and then concatenated for classification via an MLP. These strategies yielded accuracy scores of 69.77% and 70.25%, respectively, indicating that straightforward feature fusion is insufficient for modeling the complex dependencies present in brain connectome data. In contrast, more sophisticated multimodal models—including GCNN [9], Joint GCN [10], Joint DCCA [11], and BrainNN [12]—delivered higher performance, with accuracy values ranging from 70.07% to 77.45%. BrainNN [12], leveraging graph contrastive learning based fusion mechanisms, achieved the strongest baseline accuracy at 77.45%, highlighting the benefit of advanced cross-modal integration.

Our proposed NeuroKoop framework achieved the highest overall accuracy (82.33%), representing an improvement of

TABLE I
COMPARISON AGAINST SOTA APPROACHES [UNIT: %] (MEAN ± STANDARD DEVIATION).

| Method | Accuracy | Precision | Recall | F1-score |
|---|---|---|---|---|
| Logistic Regression | 66.82 ± 1.3152 | 66.51 ± 1.2661 | 66.40 ± 1.3351 | 66.60 ± 1.4881 |
| Random Forest | 68.94 ± 1.2279 | 70.01 ± 1.6691 | 68.62 ± 1.7522 | 68.85 ± 1.5792 |
| GAT [24] | 69.77 ± 0.0011 | 70.89 ± 0.2436 | 69.40 ± 0.4899 | 69.57 ± 0.3254 |
| Graph Transformer [25] | 70.25 ± 0.0335 | 71.12 ± 0.2228 | 70.67 ± 0.2295 | 70.26 ± 0.1428 |
| GCNN [9] | 70.07 ± 0.1589 | 70.82 ± 0.2282 | 70.93 ± 0.2148 | 69.92 ± 0.1610 |
| Joint GCN [10] | 74.53 ± 0.0186 | 75.66 ± 0.0261 | 74.65 ± 0.0213 | 74.91 ± 0.0228 |
| Joint DCCA [11] | 75.67 ± 0.1302 | 75.29 ± 0.2162 | 75.89 ± 0.2054 | 75.33 ± 0.2667 |
| BrainNN [12] | 77.45 ± 0.0212 | 77.89 ± 0.0252 | 77.09 ± 0.0213 | 77.18 ± 0.0237 |
| **NeuroKoop [Proposed]** | **82.33 ± 0.0197** | **82.49 ± 0.0178** | **82.09 ± 0.0342** | **82.26 ± 0.0215** |

TABLE II
COMPARISON OF PERFORMANCES WITH FIVE DISTINCT RANDOM SUBSETS FROM UNEXPOSED COHORT [UNIT: %] (MEAN ± STANDARD DEVIATION).

| Trial | Accuracy | Precision | Recall | F1-score |
|---|---|---|---|---|
| Subset 1 | 82.12 ± 0.0189 | 82.23 ± 0.0189 | 81.88 ± 0.0337 | 82.04 ± 0.0268 |
| **Subset 2** | **82.33 ± 0.0197** | **82.49 ± 0.0178** | **82.09 ± 0.0342** | **82.26 ± 0.0215** |
| Subset 3 | 82.17 ± 0.0191 | 82.32 ± 0.0181 | 81.97 ± 0.0392 | 82.14 ± 0.0244 |
| Subset 4 | 82.08 ± 0.0201 | 82.20 ± 0.0212 | 81.84 ± 0.0289 | 82.00 ± 0.0291 |
| Subset 5 | 82.26 ± 0.0185 | 82.39 ± 0.0280 | 82.01 ± 0.0311 | 82.19 ± 0.0223 |

TABLE III
ABLATION EXPERIMENT OUTCOMES [UNIT: %] (MEAN ± STANDARD DEVIATION).

| NeuroKoop variants | Accuracy | Precision | Recall | F1-score |
|---|---|---|---|---|
| w/o WM Scores ($c$) | 80.16 ± 0.0141 | 80.07 ± 0.0160 | 79.77 ± 0.0119 | 80.90 ± 0.0138 |
| w/o Cross-modal Attention | 75.47 ± 0.0306 | 74.35 ± 0.0339 | 74.91 ± 0.0337 | 75.05 ± 0.0295 |
| w/o Neural Koopman Layer | 70.12 ± 0.0267 | 71.70 ± 0.0367 | 71.86 ± 0.0324 | 70.65 ± 0.0200 |
| w/o Adversarial Loss ($\mathcal{L}_{adv}$) | 80.51 ± 0.0256 | 79.02 ± 0.0340 | 79.56 ± 0.0345 | 80.71 ± 0.0240 |
| FNC only + Neural Koopman Layer | 76.98 ± 0.0587 | 77.25 ± 0.0429 | 76.73 ± 0.0821 | 76.82 ± 0.0437 |
| SBM only + Neural Koopman Layer | 74.61 ± 0.0362 | 76.04 ± 0.0518 | 74.25 ± 0.0778 | 74.41 ± 0.0395 |
| **Proposed** | **82.33 ± 0.0197** | **82.49 ± 0.0178** | **82.09 ± 0.0342** | **82.26 ± 0.0215** |

nearly five percentage points over the best competing method. This margin was consistently maintained across validation folds, as reflected in the low standard deviation. These results underscore the effectiveness of our approach in extracting and integrating complementary patterns from both connectome modalities.

To further assess NeuroKoop's robustness, we conducted an additional analysis using five randomly drawn subsets of 430 unexposed subjects of ABCD baseline cohort, each paired with the same set of 430 exposed individuals. As shown in Table II, NeuroKoop achieved consistently high and stable performance across all five runs, with minimal variation across accuracy, precision, recall, and F1-score. Among these, Subset 2 yielded the highest performance and is the one reported and discussed in this paper for comparative analysis (Table I, III). These outcomes confirm that the model's predictive capacity is not sensitive to the specific control group composition, reinforcing the general reliability and robustness of our findings.

*2) Ablation Experiments:* To assess the contribution of each architectural component, we performed a series of ablation studies, systematically removing key modules, investigating different configurations and measuring the impact on classification performance, as reported in Table III. We addressed five guiding research questions (RQs): (RQ1) What is the contribution of WM scores? (RQ2) How essential is cross-modal attention? (RQ3) What is the effect of the neural Koopman layer? (RQ4) Does adversarial loss enhance model generalization? (RQ5) How does the full multimodal fusion compare with unimodal Koopman configurations?

Removing WM scores reduced accuracy to 80.16%, highlighting their benefit for PDE status detection. Omitting cross-modal attention led to a pronounced drop to 75.47%, underscoring its importance for modeling inter-modality relationships. Excluding the Koopman layer produced the largest decline (70.12%), indicating its crucial role in capturing network dynamics. The removal of adversarial loss resulted in a moderate decrease to 80.51%, suggesting its utility for generalization. Finally, by applying the Koopman operator to fMRI-derived FNC and sMRI-derived SBM graphs separately, we further evaluated two unimodal configurations. Both setups underperformed compared to the full NeuroKoop framework, with FNC+Koopman achieving 76.98% accuracy and SBM+Koopman reaching 74.61%. These results indicate that while the Koopman mechanism improves single-modality modeling, fusing both modalities within a unified latent space leads to significantly stronger performance across all metrics. This supports our central hypothesis that dynamic integration of structural and functional information offers a more

comprehensive representation of PDE-related brain alterations. Collectively, these findings demonstrate that each component makes a unique and essential contribution to the overall model performance.

### D. Qualitative Evaluation

To elucidate the neural pathways most relevant for PDE classification, we analyzed the top 3% of cross-modal attention weights from NeuroKoop's structural-to-functional (SBM-to-FNC) attention layer as illustrated in Figure 2. These weights quantify how each structural network draws upon information from functional networks during multimodal integration. By averaging attention patterns within each group, we highlighted the connections most prominent in refining the fused representation.

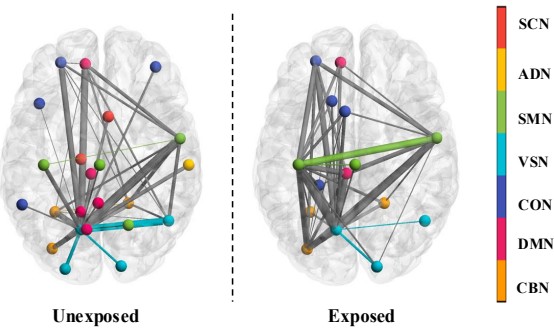

Fig. 2. Axial depiction of group-averaged cross-modal attention, showing the top 3% strongest connections for both unexposed and exposed groups across seven brain networks from *NeuroMark*—subcortical (SCN), auditory (ADN), sensorimotor (SMN), visual (VSN), cognitive control (CON), default mode (DMN), and cerebellar (CBN) network. Edges within the same network are color-coded, while those linking different networks are shown in gray. Edge thickness is proportional to attention weight, highlighting the relative importance of each connection in the fused representation.

Among unexposed individuals, the highest cross-modal attention was observed between the sensorimotor (SMN), visual (VSN), cognitive control (CON) and default mode (DMN) networks, consistent with broadly distributed integration typical of normative brain development. In contrast, the exposed group exhibits more concentrated attention patterns, with the majority of high-weighted connections clustered among the DMN, CON, and cerebellar (CBN) networks. This focused pattern aligns with recent studies [29]–[32] showing that that PDE is associated with associated with reorganization and tighter coupling within DMN, CON, and CBN, possibly reflecting compensatory or maladaptive changes in neural communication. Such focused connectivity within the exposed group indicates a shift toward selective engagement of key brain networks, potentially resulting from PDE.

## IV. CONCLUSIONS

We introduced NeuroKoop, a novel GNN based multimodal framework that leverages Koopman operator theory to fuse multimodal connectomes for robust classification of PDE status. By integrating WM score as a personalized control input, our approach enabled dynamic latent space fusion, capturing subtle structure–function relationships beyond conventional fusion approaches. Applied to ABCD dataset, NeuroKoop consistently outperformed existing SOTA multimodal baselines, highlighting its potential to advance individualized risk assessment and mechanistic understanding in neurodevelopmental research. Future work will explore more expressive neural operators and self-supervised strategies to further enhance generalizability and interpretability for multimodal brain network analysis.

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
