# OpenReview forum: "NeuroKoop: Neural Koopman Fusion of Structural–Functional Connectomes for Identifying Prenatal Drug Exposure in Adolescents"
_IEEE.org/EMBS/BHI/2025/Conference — BHI 2025_

### Official Review · Reviewer_XG9a · 2025-07-15
**Promising Method with Concerns on Evaluation and Model Design**

**Confidence:** 4
**Clarity Of Writing:** good
**Clinical Significance:** good
**Methodological Novelty:** good
**Overall Rating:** 6

**Experiments And Results:**

fair

**Questions For The Authors:**

NA

**Strengths:**

The manuscript is well-written and generally easy to follow.
The proposed approach is interesting and shows innovation in combining multimodal neuroimaging with cognitive scores.

**Summary Of The Paper:**

This paper aims to identify prenatal drug exposure using fMRI and sMRI data, combined with working memory (WM) scores. To improve performance, the study proposes a cross-modal fusion of the two imaging modalities and uses the WM score as additional information injected into the model.

**Weaknesses:**

1.	The description of the dataset and data preprocessing (Section III-A) should not be included in the Results and Discussion section.
2.	In Section III-A, the dataset is downsampled to create a balanced comparison. While this may be reasonable for the training set, balancing the test set changes the real-world data distribution. This raises concerns about whether the reported performance reflects true effectiveness under real-world, imbalanced conditions.
3.	The paper does not clearly demonstrate the advantage of the proposed cross-modal fusion over more direct or standard fusion techniques. Additional experiments or analysis comparing these approaches would strengthen the contribution.
4.	In the introduction, the meaning of the “second obstacle” is unclear and should be explained more precisely.
5.	In the model architecture, only a single feature (c, the WM score) is used as input for the MLP (M). It is unclear how the MLP can extract meaningful information from just one feature.

---

### Official Review · Reviewer_wc1X · 2025-07-15
**Promising multimodal connectome fusion for detecting prenatal drug exposure with a novel methodology**

**Confidence:** 4
**Clarity Of Writing:** good
**Clinical Significance:** great
**Methodological Novelty:** great
**Overall Rating:** 7

**Experiments And Results:**

good

**Questions For The Authors:**

- Can the authors comment on how stable the results are to the choice of Koopman iteration number and WM-conditioning weight?
- How does the proposed method compare with a much simpler logistic-regression or random-forest classifier built on global SBM + FNC summary metrics? A strong performance gap would strengthen the case for architectural complexity.

**Strengths:**

- This paper tackles an impactful public-health question of how prenatal drug exposure alters adolescent brain organization.
- Methodological novelty: This paper introduces a neural-Koopman operator that couples structural and functional graphs while conditioning on individual WM performance, moving beyond simple feature concatenation, or conventional multimodal fusion.
- Interpretability: Attention map analysis highlights specific network-level alterations, giving neuroscientists actionable leads.

**Summary Of The Paper:**

This paper addresses how prenatal drug exposure (PDE) may alter the developing brain by introducing NeuroKoop, a framework that fuses structural and functional MRI information from adolescents in the ABCD study. Instead of treating each imaging modality separately, the model learns a shared representation of brain organization. It also factors in each participant’s working-memory performance. When evaluated on the task of distinguishing 430 PDE youths from 430 matched controls, the proposed method outperforms several standard unimodal and multimodal baselines on various classification metrics.

**Weaknesses:**

- Limited external validation: The paper evaluates performance on a single ABCD subset; generalizability to other cohorts or scanner protocols remains unknown.
- Model complexity vs. sample size: The large multi-component architecture may risk over-fitting given only the small dataset size (~430 cases). Learning-curve or variance analyses are absent.
- Baseline comparability: Competing multimodal methods were implemented by the authors, but hyper-parameter selection criteria and training budgets are not harmonized, making the 5% accuracy gains difficult to interpret.

---

### Official Review · Reviewer_fRoR · 2025-07-18
**A multimodal GNN framework**

**Confidence:** 4
**Clarity Of Writing:** great
**Clinical Significance:** great
**Methodological Novelty:** great
**Overall Rating:** 6
**Final Rating:** 7

**Experiments And Results:**

great

**Questions For The Authors:**

1. How did the model perform with different numbers of Koopman evolution steps? Is it sensitive to this setting?
2. How was the value of k chosen for constructing the k-nearest neighbor graphs? Would a dynamic or learned k improve performance?

**Strengths:**

1. The use of a neural Koopman operator to refine brain network allows the model to simulate complex brain dynamics more effectively than traditional fusion methods. And including working memory scores as extra input helps the model account for individual cognitive differences.
2. The paper clearly explains how data is processed and how each part of the model works, and instead of simply combining features from structural and functional brain scans, the model merges them in a deep and structured way.
3. NeuroKoop consistently outperforms other SOTA models and ablation experiments show the importance of each part of the model.

**Summary Of The Paper:**

This paper presents NeuroKoop, a novel multimodal GNN framework designed to classify prenatal drug exposure (PDE) in adolescents. The authors use the ABCD dataset and fuse structural brain data and functional brain data to build subject-specific brain graphs. The model outperforms state-of-the-art baselines, and ablation studies validate the contribution of each module.

**Weaknesses:**

1. While the attention weights offer some understanding of model behavior, additional interpretability methods (like saliency maps or node importance scores) could help explain exactly what drives each prediction.
2. The paper doesn’t explain how cases with missing WM scores or incomplete imaging were managed.

---

### Official Review · Reviewer_tJaV · 2025-07-20
**Koopman operator-based method for fmri & smri fusion learning**

**Confidence:** 5
**Clarity Of Writing:** great
**Clinical Significance:** good
**Methodological Novelty:** great
**Overall Rating:** 7

**Experiments And Results:**

great

**Questions For The Authors:**

Can further attempt ablation experiments of the Koopman operator on single modalities.

**Strengths:**

1. The paper proposes a Koopman operator that effectively solves the mapping of nonlinear dynamical systems to linear spaces.
2. From the results and ablation experiments, the fusion of fMRI and sMRI shows certain effectiveness.
3. Considering discrimination as a way to enhance inter-modal distribution alignment is quite an interesting perspective.

**Summary Of The Paper:**

NeuroKoop is a graph neural network framework that fuses structural and functional brain connectomes using neural Koopman operator theory to identify prenatal drug exposure in adolescents. Applied to 860 participants from the ABCD dataset, the method extracts 53 brain networks via ICA, uses cross-modal attention and working memory scores for dynamic fusion, and achieves 82.33% accuracy in classifying exposure status, outperforming existing baselines while revealing that exposed individuals exhibit more concentrated connectivity patterns in default mode, cognitive control, and cerebellar networks.

**Weaknesses:**

The overall framework design is reasonable and has certain feasibility, but I have several concerns:

1. It is currently unclear why working memory scores were chosen to be integrated into the neural network as "auxiliary information," but this lacks sufficient neuroscientific theoretical support. I suggest supplementing relevant literature. The rationale for why working memory is particularly important, rather than other cognitive functions, has not been explained.

2. I am curious whether the size of the graph established by k-nn affects the overall performance of the downstream task for the subsequent model. I suggest conducting ablation studies to select the most appropriate k value.

3. Given the obvious sample imbalance in the dataset, I suggest trying random selection of negative samples to see if the model is also robust under these conditions.

4. One aspect I'm concerned about is whether using Koopman operator layers for single-modality fMRI would perform better than incorporating sMRI, because sMRI has stronger structural characteristics and it's not easy to solve fMRI's dynamic features through fusion alone. This requires further ablation experiments to be proposed.